# Metformin Prevents or Delays the Development and Progression of Osteoarthritis: New Insight and Mechanism of Action

**DOI:** 10.3390/cells11193012

**Published:** 2022-09-27

**Authors:** Miao He, Bangbao Lu, Michael Opoku, Liang Zhang, Wenqing Xie, Hongfu Jin, Siyu Chen, Yusheng Li, Zhenhan Deng

**Affiliations:** 1Department of Orthopaedics, Xiangya Hospital, Central South University, Changsha 410008, China; 2Department of Sports Medicine, The First Affiliated Hospital of Shenzhen University, Shenzhen Second People’s Hospital, Shenzhen 518035, China; 3National Clinical Research Center for Geriatric Disorders, Xiangya Hospital, Central South University, Changsha 410008, China

**Keywords:** metformin, osteoarthritis, type 2 diabetes, AMPK, SIRT1

## Abstract

For over 60 years, metformin has been widely prescribed by physicians to treat type 2 diabetes. Along with more in-depth research on metformin and its molecular mechanism in recent decades, metformin has also been proposed as an effective drug to prevent or delay musculoskeletal disorders, including osteoarthritis (OA). The occurrence and development of OA are deemed to be associated with the impaired mitochondrial functions of articular chondrocytes. Metformin can activate the pathways and expressions of both AMPK and SIRT1 so as to protect the mitochondrial function of chondrocytes, thereby promoting osteoblast production. Moreover, the clinical significance of the metformin combination therapy in preventing OA has also been demonstrated. This review aimed to comprehensively summarize the current research progress on metformin as a proposed drug for OA prevention or treatment.

## 1. Introduction

Metformin, a biguanide, is considered to be a classic drug for the treatment of type 2 diabetes (T2D) (hypoglycemic effect) and has been clinically used for over 60 years [1]. Among various anti-diabetic drugs, metformin can effectively treat diabetics with few adverse effects [2,3]. Along with more in-depth research on metformin in the past decades, its therapeutic effect has also been expanded to other diseases, such as cardiovascular disease [4,5], liver disease [6,7], obesity [8], polycystic ovary syndrome (PCOS) [9,10], tumor [11], aging [12,13,14,15], and osteoporosis [16]. Recently, it was reported that metformin administered shortly after joint injury can limit osteoarthritis (OA) development and progression in injury-induced OA animal models [17].

OA is a chronic, slowly progressive, degenerative joint disease affecting the musculoskeletal system, particularly in the elderly patients. So far, there has been no specific or effective approach to reversing the progression of OA; however, conservative treatment supported by non-steroidal anti-inflammatory drugs (NSAISDs), regular exercise, physiotherapy, modification of lifestyle and body weight control has been demonstrated helpful [18]. Most of the elderly patients in the late stage of OA can only relieve their symptoms by surgical treatment. According to the latest epidemiological statistics, the number of OA patients worldwide had reached 303 million in 2017 and is expected to continue growing, especially in countries facing the aging problem [19,20]. Thus, there is an urgent need to find effective drugs to inhibit the occurrence and progression of OA. The OA incidence, disease type, location, and the number of affected joints are believed to associate with race, age, occupation, lifestyle, and genetic factors. OA may involve structural changes of the articular cartilage, subchondral bone, ligament, joint capsule, synovium, muscles around joints [21], and a multitude of further factors (including genetic, biomedical, and mechanical factors) have been implicated throughout its development [18].

The anti-diabetic effect of metformin is mediated by the activation of adenosine monophosphate (AMP)-activated protein kinase (AMPK) due to the blockade of the mitochondria respiratory chain, which results in an increased AMP/adenosine triphosphate (ATP) ratio [5]. The AMPK signaling pathway had been proven to be involved in the cartilage metabolism of OA. Decreased AMPK activity has been observed in the chondrocytes and articular cartilage tissue from OA patients [22]. It was also reported that the specific knockout (KO) of the mammalian target of rapamycin (mTOR) in chondrocytes could upregulate AMPK and increase autophagy, thereby preventing surgically induced OA in mice [23]. AMPKα1α2 conditional double KO (AMPKα cDKO) mice showed severe and spontaneous age-associated OA symptoms and an enhanced IL-1β-stimulated catabolic response, suggesting that AMPK activity in chondrocytes is important for maintenance of joint homeostasis [24]. The treatment of AMPKα1 KO and the destabilization of the medial meniscus (DMM)-induced OA mice with metformin had no effect, suggesting that the chondro-protective effect of metformin was mediated mainly by the up-regulation of AMPKα1 expression [25]. This review aimed to summarize the latest research progress of metformin in relation to OA and to elucidate its specific mechanism in controlling OA occurrence and development. These findings may provide a new insight into the metformin mechanisms in OA prevention.

## 2. The History, Characteristics, and Mechanism of Metformin

In the late 19th century, guanidine was successfully extracted from G. officinalis and was used in medieval Europe to relieve polyuria in diabetic patients [26,27]. Metformin was first synthesized in 1922, but its role was long ignored due to the synthesis of insulin almost around the same time. It was until 1957 that metformin became known to the public as a therapeutic agent for diabetes. Metformin was brought into the United States in 1995, and in 2003, it was listed as the essential drug for diabetes by the World Health Organization (WHO) [28].

Metformin (N, N-dimethylbiguanide) consists of planar molecules with single protonation between two imino groups and two non-polar methyl groups, and its molecular formula is C4H11N5 (molecular weight 129.16 g/mol). Instead of being extracted from plants, clinically used metformin (metformin hydrochloride, C4H11N5•HCL, with a molecular weight of 165.6 g/mol) is synthesized from chlorinated dimethylamine and dicyandiamide. Metformin hydrochloride presents in the form of white powder at room temperature. Its melting point is 223–226 °C, and its solubility in water is 200 g/L [29].

T2D is caused by insulin resistance and is related to obesity. Metformin can inhibit the absorption of glucose by intestinal parietal cells and block the glycogenolysic and metabolic pathways to achieve a hypoglycemic effect [30]. In short, metformin works by blocking the mitochondrial respiratory chain complex 1 activity, thereby inducing the separation of oxidative phosphorylation and increasing the AMP/ATP ratio. Oxidative phosphorylation is the process by which ATP synthesis is coupled to the movement of electrons through the mitochondrial electron transport chain and the associated consumption of oxygen [31,32].

Mitochondrial injury is concerned with the associations of the pathological process of OA. It was reported that anti-diabetic drugs (i.e., metformin) could activate the Sirtuin (SIRT) 3 expression to protect mitochondria from oxidative stress. Wang et al. conducted a study to examine the inhibitory property of metformin on mitochondrial damage by focusing on the interleukin-1 beta (IL-1β)-stimulated OA model by using primary murine chondrocytes. The results showed that SIRT3 was downregulated in chondrocytes under IL-1β stimulation, and its expression was related to the mitochondrial injury and the production of reactive oxygen species (ROS). Metformin treatment could upregulate the expression of SIRT3, mitigate the loss of cell viability, and decrease the generation of mitochondria-induced ROS in chondrocytes aroused with IL-1β [33].

Metformin has the advantages of low price, high safety in application of T2D treatment, and it rarely causes obvious hypoglycemia [34,35]. It can be used either alone or in combination with other hypoglycemic drugs [36] and exhibits prolonged safety properties in different categories of patients with or without diabetes [37]. Today, metformin is widely prescribed all over the world, and its application is no longer confined to diabetes.

## 3. Mechanism of Metformin in Preventing OA Development and Progression

How metformin interferes with the development and progression of OA is very complicated (Figure 1). The current research shows that its mechanism mainly involves the activation of the AMPK signaling pathway, the SIRT1 protein regulator, as well as the combination therapy with other medications (Figure 2, Table 1).

Metformin plays its potential beneficial role in a variety of diseases mainly through its mitochondrial effect [37]. Owen et al. first pointed out that metformin’s primary site of action is through a direct inhibition of complex 1 of the respiratory chain during treatment of non-insulin-dependent diabetes [50]. In addition, EI-Mirr et al. confirmed that metformin could slowly penetrate the mitochondrial inner membrane and directly inhibit the mitochondrial respiratory chain complex 1 [51]. Metformin mainly acts on mitochondria via the cell membrane organic cation transporters (OCT), so as to inhibit the mitochondrial respiratory chain complex I (which is allosterically activated). Both AMPK and activated AMPK are able to switch cells from the anabolic state to catabolic state, disable the synthetic pathways that consume ATP, and restore the energy balance [32,35]. AMPK is a key regulator of metabolism because it senses increases in the intracellular ratio of AMP and/or adenosine diphosphate (ADP) to ATP following cellular stress, which then triggers a metabolic switch from ATP consumption to ATP generation to maintain energy balance [52]. During signal activation, three kinases/phosphatases, namely, Liver kinase B1 (LKB1), calcium/calmodulin-dependent protein kinase II (CaMKII), and transforming growth factor-β-activated kinase 1 (TAK1), may be involved in regulating AMPK [53,54,55,56]. Metformin can directly act on AMPK and SIRT1 or play a role via a certain link between them. These mechanisms are specifically described in detail below.

### 3.1. Metformin Affects the Formation of OA via AMPK Activation

Currently available studies have proven that the activation of AMPK can exert a great effect on musculoskeletal disorders. AMPK, as a regulatory factor, has been deemed a targeted option for OA inhibition [57].

Decreased AMPK activity was found in human samples as well as mice OA knee joint chondrocytes, indicating that the AMPK activity may protect the articular cartilage from degeneration [22,51]. After observing mice administered with metformin for specified time periods, it was found that metformin was able to increase AMPK activity significantly. Compared with the control group (without administration of metformin), a significant improvement was detected in the Osteoarthritis Research Society International (OARSI) scores, proving that metformin can mediate the AMPK activity and thereby inhibit OA advancement [17]. Hui Li et al. [38] constructed a mouse model of OA through the DMM surgery and then injected saline or metformin into the stomach and knee joint cavity of the mice, respectively. The degeneration of the knee cartilage with greatly lowered OARSI scores was observed in the intragastric and intra-articular metformin samples as compared to those injected with intragastric/intra-articular saline. The investigators further treated the chondrocytes with 10 mmol metformin for 24 h in vitro and found that the expression of phosphorylated Ampkα (pAMPKα), but not AMPKα1, in IL-1β-cultured chondrocytes was changed significantly.

AMPK plays important roles in osteoblast differentiation and mineralization. Kanazawa et al. found that osteoblastic AMPK plays important roles in bone development in vivo and that the deletion of AMPK in osteoblasts decreases osteoblastic differentiation and enhances bone turnover by increasing receptor activator of nuclear factor κB ligand (RANKL) expression [58]. Mai et al. found that inhibition of AMPK and CaM kinase kinase (CaMKK), two targets of metformin, suppressed endogenous and metformin-induced osteoprotegerin (OPG) secretion in osteoblasts. Metformin is able to reduce RANKL and stimulates OPG expression in osteoblasts, further inhibit osteoclast differentiation and prevent bone loss [59]. As osteoblast is also enrolled in OA progression, metformin might prevent OA progression through the similar pathway.

The KO of AMPK α or β subunits was found to lead to a reduction of trabecular bone density and quality in transgenic animal studies [60]. It was reported that AMPK could stimulate the osteogenesis of MC3T3-E1 cells via the AMPKGfi1–osteopontin (OPN) axis pathway [61]. Further, the runt-related transcription factor 2 (RUNX2) and Wnt/β-catenin pathways can regulate the differentiation of bone mesenchymal stem cells (MSCs) to osteoblasts [62,63]. By injecting metformin combined with alendronate into the collagenase-induced mouse OA model, the results showed that metformin inhibited the expressions of both the receptor activator of nuclear factor kappa-B (RANK) and RANK ligand (RANKL) on osteoblasts and osteoclasts. In addition, it exerted a suppressive effect on the differentiation of both MSCs and fibroblasts and decreased the serum concentrations of leptin and resistin in the chronic phase of arthritis [39].

The RANKL protein has been considered as an essential biomolecule in the processes of osteoclast activation and proliferation, while the AMPK signaling pathway can downregulate the expressions of peroxisome proliferator-activated receptor γ (PPARγ)-1, nuclear factor of activated T cell 1 (NFATC1), parathyroid hormone-related protein (PTHrP), and mevalonate, thereby leading to the inhibition of RANKL. Subsequently, this process will further inhibit osteoclastogenesis. The expression of PPARγ-2 can suppress osteoblast differentiation and induce MSCs to differentiate into adipocytes [64]. The mevalonate pathway, which plays a role in the prenylation of regulative proteins like Ras and Rho GTPase, exerts a negative effect on bone tissue [65]. The specific deletion of AMPKa1 in murine arthritis may lead to the compensatory upregulation of AMPKα2 [24], and the silencing of AMPKα2 can prevent the damage of chondrocytes, implying that AMPKα2 plays a major role in AMPK regulation [66].

The AMPK/mTORC1 (a mammalian target of rapamycin complex 1) signaling pathway has been proven to associate with the progression of OA [67]. The gavage of rapamycin to the surgically induced OA mouse model could specifically inhibit the activity of the mTORC1 pathway of articular chondrocytes, promote autophagy, and maintain the survival of articular chondrocytes, thus delaying the onset of OA [68]. Lower OARSI scores and fewer matrix metalloproteinase (MMP)-13-positive cells were detected in DMM mice and cartilage explants after treatment with metformin. Additionally, metformin is able to decrease the p16INK4a level in OA chondrocytes and enhance the polarization of AMPK and inhibit mTORC1 in OA mice and chondrocytes in a dose-dependent manner [40]. The chondrocyte-specific KO of mTOR was reported to upregulate AMPK, increase autophagy, and prevent the occurrence of surgery-induced OA in mice [23]. Zhang et al. found that metformin was able to increase cell proliferation, alleviate the IL-1β-induced extracellular matrix (ECM) metabolic imbalance and proinflammatory cytokine production, and exert an anti-apoptosis activity in murine OA chondrocytes, possibly by regulating the AMPK/nuclear factor κB (NF-κB) signaling pathway [41].

The anti-inflammatory benefits of metformin is another mechanism of OA prevention. Ye et al. found that metformin is able to inhibit the phosphorylation of I-κBα and p65 while it activates AMPK, suggesting that metformin inhibits LPS-induced chemokine expression through the AMPK and NF-κB signaling pathways [69]. Meanwhile, metformin can regulate the expression of M1/M2 and induce macrophage polarization through the AMPK activity and can also increase the expressions of some cytokines [70]. Long-term treatment with metformin was reported to increase C-JNK-1 phosphorylation, reduce fat deposition, and enhance AMPK phosphorylation [71]. Thus, by inducing cell autophagy through AMPK activation, metformin may eventually prevent OA. In summary, it can be concluded that metformin is capable of regulating AMPK, inhibiting mTORC1, enhancing autophagy, and thereby alleviating or preventing OA occurrence.

### 3.2. Metformin Protects Cartilage by Activating SIRT1 Protein Regulator

The sirtuins family is a group of well-known anti-aging genes [72]. The recent literature suggests that SIRT1 is associated with a range of age-related diseases, including OA [73]. SIRT1 is one of the major metabolic proteases in different metabolic tissues [74]. It exerts similar functions to AMPK in metabolism and cell survival, and plays a crucial role in tissue homeostasis, including cartilage. AMPK and SIRT1 each regulate one another and share several common target molecules. A growing amount of evidence suggests that these two molecules have similar effects on multiple processes, such as cellular fuel metabolism, inflammation, and mitochondrial function. The research on the partnership between AMPK and SIRT1 is long-standing, providing meaningful implications for our understanding of the pathologic process and medical care of disorders related to metabolic syndromes and aging [75].

In the hypothetical SIRT1/AMPK cycle, a decrease in energy state or the activation of AMPK by other means is deemed to lead to activation of SIRT1, perhaps by increasing nicotinamide adenine dinucleotide (NAD+) or the NAD/NADH ratio and/or the activity of Nampt [76,77]. SIRT1 then deacetylates and activates LKB1, which successively activates AMPK. As an alternative, these events may also be set in motion by factors that primarily increase SIRT1. The joint activation of SIRT1 and AMPK permits for the simultaneous deacetylation and phosphorylation of the listed target molecules and presumptively others.

A lowered expression of SIRT1 protein is often associated with OA. Several animal experiments have shown that the progression of OA in mice with KO of cartilage-specific SIRT1 may be accelerated after mechanical stress [78]. The SIRT1 downstream gene p53 and its acetylation are massively elevated in patients with knee OA and are correlated with the severity of the disease [79].

In addition to above-mentioned statements, metformin also exerts a protective effect on cell senescence by inducing the SIRT1 expression [80]. Metformin was applied to DMM induced OA mice knee and IL-1β-induced chondrocytes. Metformin increased the phosphorylated levels of AMPKα and upregulated SIRT1 protein expression, leading to an increase in autophagy as well as a decrease in catabolism and apoptosis. Deactivating AMPKα or inhibiting SIRT1 prevented the augmented autophagy in the presence of metformin. Silencing AMPKα2, but not AMPKα1, reduced SIRT1 expression and downregulated autophagy in cultured chondrocytes, suggesting the chondroprotective effect of metformin was through activating AMPKα/SIRT1 signaling pathway [42]. Several downstream kinases and IGF receptors can be triggered by SIRT1 (i.e., by enhancing IGFs signaling), which can lead to the phosphorylation of murine double minute 2 (MDM2) and the blockage of the apoptotic pathway, thereby affecting the viability of chondrocytes [81]. SIRT1 regulates the expressions of several genes that are closely related to the onset and the pathological process of OA. It has been well known that these corresponding genes, such as MMP-1, 2, 9, 13, ADAMTS-5, etc., are inhibited by SIRT1 stimulation to delay OA development [82]. The increased activity of SIRT1 protein can then greatly elevate other gene expressions.

The accumulation of the activations of the receptor for advanced glycation end-products (RAGE) in chondrocytes can stimulate the production of MMP-13 [83]. RAGE is known to impose a great risk on the onset and development of OA. Metformin can induce the expressions of AMPK and SIRT1 protein and inhibit the formation of inflammatory factors as well as protein glycation. When incubated with globin-free bovine serum albumin, metformin could significantly inhibit the characteristic fluorescence of RAGE [84]. The intracellular antioxidant properties of metformin were able to inhibit lectin-like oxidation 1 (LOX-1) and RAGE through the activation of advanced glycation end products (AGE) [85].

Thus, the metformin-induced SIRT1 activation can reduce inflammatory mediators and matrix degradation substances, inhibit the accumulation of RAGE, protect the articular cartilage, and, consequently, prevent the occurrence of OA.

### 3.3. Metformin and Metal Ions Cross-Talk with the Human Serum Albumin

Human serum albumin (HSA) is the main protein that is used for examining drug viability in in vitro studies [86]. HSA displays multiple ligand binding sites with extraordinary binding capacities for a wide range of ions and molecules. It is currently an effective clinical tool for drug delivery and a potential contender as molecular cargo and nano-vehicle used in clinical practice [87]. The interaction between metformin and HSA, as well as the glycated form of HSA (gHSA), has been investigated by multiple methods, including spectroscopic techniques, zeta potential, and molecular modeling under physiological conditions. By conducting site marker competitive experiments and molecular modeling, the binding site of the drug to the proteins was able to be exactly located, and the result showed that metformin was located at subdomain IIIA on HSA and gHSA, respectively [88]. This is of great importance to the determination of drug dosage in disease treatment.

HSA is also the major carrier of metal ions, such as zinc (II) and copper (II), in the blood plasma [89]. In the kidney of T2D patients, the toxic effect of copper (II)/amylin adducts on cells will be enhanced by the presence of metformin and the successive formation of the ternary copper (II)/amylin/metformin complex, which may be correlated with the diabetic nephropathy development [90]. Metformin-treated diabetic patients have a lower cancer incidence. Metformin forms stable complexes with copper (II) ions and exerts an anti-proliferative effect on tumor cells, which can be enhanced in the presence of copper ions [91]. OA shares some common pathologies with these diseases and revealing the metformin and metal ion interactions might be the key controlling factor for OA treatment strategy.

### 3.4. Metformin Attenuates Cartilage Degeneration and OA Progression

Both the single and combined administrations of metformin are effective in attenuating cartilage degeneration and OA progression. The expression of collagen type II (Col II) was increased, while MMP-13, NOD-like receptor protein 3 (NLRP3), caspase-1, gasdermin D (GSDMD), and IL-1β was significantly deceased at 4 or 8 weeks after metformin injection into DMM-induced OA mice. Meanwhile, the OARSI scores and the thickness of calcified cartilage were decreased while the thickness of hyaline cartilage was increased [43].

The combined administration of metformin and celecoxib can control cartilage damage more effectively than metformin alone. It can relieve pain in monosodium iodoacetate (MIA)-induced rat OA model. It was also found that this combined therapy reduced the catabolic factor gene expression and inflammatory cell death factor expression, increased the expressions of LC3Ⅱb, p62 and LAMP1, and induced an autophagy–lysosome fusion phenotype in human OA chondrocytes. These results demonstrated the potential for the therapeutic use of metformin in OA patients based on its ability to suppress pain and protect cartilage [44]. The metformin-stimulated adipose tissue-derived human MSCs (Ad-hMSCs) suppressed the expressions of RUNX2, COL X, vascular endothelial growth factor (VEGF), MMP-1, MMP-3, and MMP-13 in IL-1β-stimulated OA chondrocytes and increased the expressions of tissue inhibitor of metalloprotease 1 (TIMP1) and tissue inhibitor of metalloprotease 3 (TIMP3) in the MIA-induced rat OA model. The expression of transforming growth factor-β (TGF-β) in the subchondral bone of OA joint was attenuated more significantly in OA rats treated with metformin-stimulated Ad-hMSCs versus controls [45]. In a mouse OA model induced by estrogen deficiency and obesity, metformin administration combined with exercise training was executed, and the results indicated that increased expressions of aggrecan and Col II and decreased expressions of disintegrin and metalloproteinase with thrombospondin motifs (ADAMTS)-4 increase the concentration of osteocalcin and decrease the serum concentration of IL-1β and CTX-1 and glucose in the articular cartilage when compared with control group, suggesting that the combination of metformin administration and exercise may be a potential candidate for controlling the progression of OA [46].

## 4. Clinical Significance of Metformin Combination Therapy in OA Treatment

There is no proven method for the treatment of OA at present. The effectiveness of metformin in OA treatment by controlling or slowing down OA-related inflammation and metabolic pathways has been demonstrated by an earlier study [33]. In addition, a retrospective study on metformin in combination with a type of NSAIDs (cyclooxygenase-2 blockers) showed that OA patients with diabetes had a lower chance of undergoing total knee arthroplasty (TKA) [47]. Likewise, a lower rate of medial cartilage volume loss and a decreasing trend to undertake TKA over a 6-year period were observed in obese individuals with knee OA [48].

Meloxicam combined with metformin can significantly lower the levels of serum cytokines, thereby reducing pain and improving the joint function of radiation-induced symptomatic knee OA patients. An earlier study also showed that the sildenafil-metformin-leucine combination could reduce inflammation in vitro and even alleviate the increased expressions of alanine aminotransferase (ALT), TGF-β, IL1-β, tumor necrosis factor alpha (TNF-α), and liver collagen [92]. Moreover, compared with monotherapy, the metformin and atorvastatin combination therapy achieved a greater improvement in liver histology and lower plasma levels of oxidative stress and inflammatory markers, such as C-reactive protein (CRP), TNF-α, and IL-6 [93].

## 5. Perspective

Although metformin has shown a protective effect on cartilage through several signaling pathways, there is still a long way to go before its application in clinical practice. The first and also the most critical issue to resolve is the efficient delivery of metformin to enable the protection of cartilage from injury. Oral metformin needs to be absorbed through the digestive system and then transported through the circulatory system to finally reach the articular cartilage. Most of the metformin content may be degenerated during this process, so the dose that actually works is unknown. In addition, metformin may only have a short-term effect. Thus, the delivery of metformin within a sustained-release scaffold for local application is a modality to be further investigated. Second, local intra-articular injection of metformin may have a more direct effect with reduced adverse effects compared to systemic medication. This approach is clinically translatable with minimum difficulty and minimally invasive, as open surgery is not required. Finally, the co-delivery of metformin and anti-inflammatory factors such as IL-1 receptor antagonist and TGF-β can further enhance the role of metformin by decreasing the detrimental effects of the factors that are destructive to the cartilage. The combined application of metformin with growth factors such as bone morphogenetic proteins (BMPs), fibroblast growth factor (FGF), and platelet-derived growth factor (PDGF), as well as the micro-fracture technique, is also expected to achieve better cartilage repair outcomes. In addition, the potential adverse effects during the application of metformin, such as nausea, vomiting, stomach upset, diarrhea, weakness, and metallic taste in the mouth, should also be monitored before it becomes an official anti-OA drug.

## 6. Conclusions

As a widely known degenerative joint disease, OA has become a health threat to the whole world. Accumulated evidence has shown that metformin may have a therapeutic effect in preventing OA occurrence and development. This review summarized the recent studies on metformin as a therapeutic agent for OA, as well as on its specific mechanisms in inhibiting the onset of OA. All aspects related to the AMPK pathway, the SIRT1 expression, and the metformin combination therapy were deliberately explained based on comprehensive analysis, which concluded that metformin might be effective in preventing OA or delaying its onset.

## Figures and Tables

**Figure 1 cells-11-03012-f001:**
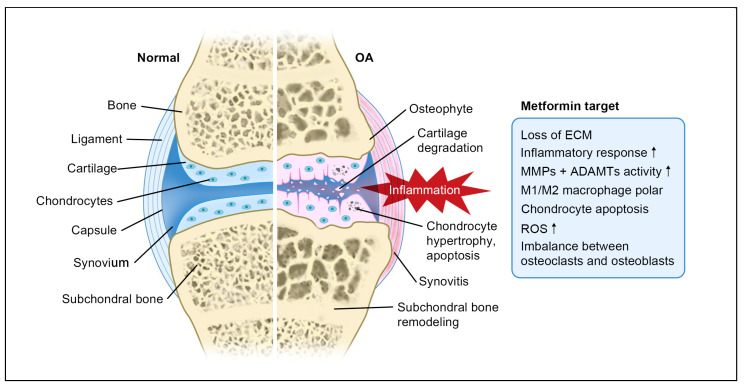
Pathogenesis of OA and the potential metformin targets. Note: ↑ means up-regulation.

**Figure 2 cells-11-03012-f002:**
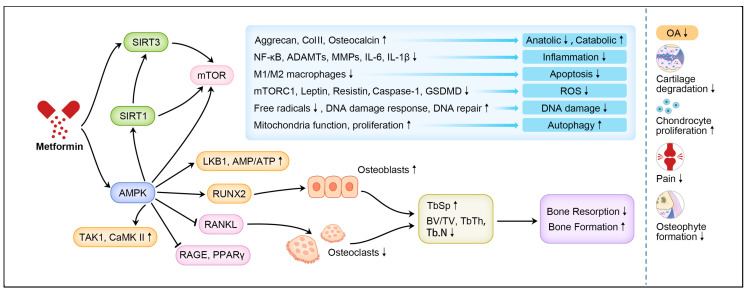
The mechanism of metformin in preventing OA. Note: ↑ means up-regulation; ↓ means down-regulation.

**Table 1 cells-11-03012-t001:** Roles of metformin in OA.

Intervention	Experimental Model	Mechanism of Action	Consequences	References
Metformin 200 mg/kg, oral, once daily for 8 weeks; metformin 0.1 mmol/kg, i.e., injection, twice a week for 8 weeks	DMM-induced mouse OA model	Decrease the level of MMP-13, but elevate Col II production by activating the AMPK pathway.	Attenuate OA structural deterioration and relieve pain.	[17]
Metformin dissolved in DMSO for in vitro assay	IL-1β-induced murine OA chondrocytes	Upregulate the expression of SIRT3; mitigate the loss of cell viability; and decrease the generation of mitochondria-induced ROS.	Suppress oxidative and OA-like inflammatory changes by enhancing the SIRT3/PINK1/Parkin signaling pathway.	[33]
Metformin 4 mg/d, oral, 2 weeks	DMM-induced mouse OA model; partial medial meniscectomy animal model of nonhuman primates	Upregulate the expressions of phosphorylated and total aMPK.	Inhibit cartilage degradation, synovial hyperplasia and osteophyte formation. Limit OA development and progression through AMPK signaling.	[38]
Metformin 10 mg/kg + alendronate 20 mg/kg, i.a. injection, 10 times (every other day from day 0 to day 18)	Collagenase-induced mouse OA model	Inhibit the expressions of RANK and RANKL on osteoblasts and osteoclasts; suppress the differentiation of both MSCs and fibroblasts; decrease the serum concentrations of leptin and resistin in the chronic phase of arthritis.	Decrease the degree of cartilage degeneration.	[39]
Metformin 100 mg/kg/d or 200 mg/kg/d, oral	DMM-induced mouse OA model	Decrease the p16^INK4a^ level in OA chondrocytes; enhance the polarization of AMPK; and inhibit mTORC1 in OA mice and chondrocytes.	Alleviate cartilage degradation and aging by regulating the AMPK/mTOR signaling pathways.	[40]
Metformin 1 mM for 24 h	IL-1β-induced murine OA chondrocytes	Increase the proliferation of chondrocytes; alleviate the IL-1β-induced ECM metabolic imbalance and proinflammatory cytokine production; exert the anti-apoptosis activity.	Protect chondrocytes by regulating the AMPK/NF-κB signaling pathway.	[41]
Metformin 1.65 g/mL, i.a. injection, once every 3 days for 8 weeks	DMM-induced mouse OA model	Restore the upregulation of MMP-13 and downregulation of Col II; increase the phosphorylated level of AMPKa and upregulate the expression of SIRT1 protein.	Increase autophagy and decrease catabolism and apoptosis by activating the AMPKa/SIRT1 signaling pathways.	[42]
Metformin 200 mg/kg/d, oral, 4 or 8 weeks	DMM-induced mouse OA model	Increase the expression of Col II and decrease the expressions of MMP-13, NLRP3, caspase-1, GSDMD and IL-1β.	Decrease the OARSI score, increase the thickness of hyaline cartilage, and decrease the thickness of calcified cartilage.	[43]
Metformin 100 mg/kg/d + celecoxib 80 mg/kg/d, oral, 14 days	MIA-induced rat OA model; chondrocytes from OA patients	Reduce the catabolic factor gene expression and the expression of inflammatory cell death factor; increase the expressions of LC3IIb, p62, and LAMP1; and induce an autophagy–lysosome fusion phenotype.	Suppress pain and protect cartilage.	[44]
Metformin-stimulated MSCs, i.v. injection	MIA-induced rat OA model	Increase the expressions of IL-10 and IDO in Ad-hMSCs and decrease the expressions of high-mobility group box 1 protein, IL-1β, and IL-6; improve the migration capacity of Ad-hMSCs by upregulating the expression of chemokine.	Exert the anti-nociceptive activity and chondroprotective effect.	[45]
Metformin 100 mg/kg/d + exercise 30 min/d, oral, 8 weeks	Estrogen deficiency and obesity induced mouse OA model	Increase the expressions of aggrecan and type II collagen (Col II) and decrease the expression of ADAMTS-4; increase the concentration of osteocalcin and decrease the serum concentrations of IL-1β, CTX-1 and glucose.	Ameliorate the abnormal metabolic status and cartilage lesions.	[46]
Metformin + COX-2 inhibitor, oral	Patients with OA and T2D	Lower the rate of receiving joint replacement surgery.	Result in a lower joint replacement surgery rate.	[47]
Metformin, oral	Obese patients with knee OA	Decrease the medial cartilage volume loss; associate with a trend towards a significant reduction of the risk of TKA.	Generate a beneficial effect on the long-term knee joint outcome.	[48]
Metformin 100 mg/kg/d, oral, 4 or 8 weeks	DMM-induced mouse OA model	Suppress the RANKL-induced activation of p-AMPK, NF-κB and pERK and the up-regulation of genes involved in osteoclastogenesis; reverse the decreases in BV/TV, Tb.Th, Tb.N, and connectivity density and the increase in Tb.Sp.	Inhibit the osteoclast formation and bone resorption in a dose-dependent manner in early OA.	[49]

Note: Ad-hMSCs, adipose tissue-derived human mesenchymal stem cells; ALT, alanine aminotransferase; AMPK, adenosine monophosphate-activated protein kinase; CTX-1, C-telopeptide of type I collagen; Col II, collagen type II; DMM, destabilization of the medial meniscus; DMSO, dimethyl sulfoxide; ECM, extracellular matrix; GSDMD, gasdermin D; IL-1β, interleukin-1β; MIA, monosodium iodoacetate; MMP-13, matrix metalloproteinase-13; mTORC1, mammalian target of rapamycin complex 1; MSCs, mesenchymal stem cells; NF-κB, nuclear factor kappa-B; NLRP3, NOD-like receptor protein 3; i.a., intra-articular; OARSI, Osteoarthritis Research Society International; pERK, phosphorylated extracellular regulated protein kinases; RAGE, receptor for advanced glycation end products. RANKL, receptor activator of nuclear factor kappa-B ligand; ROS, reactive oxygen species; SIRT, silent mating type information regulation 2 homolog; T2D, type 2 diabetes; TKA, total knee arthroplasty; IL-6, interleukin-6; CRP, C-reactive protein; TNF-α, tumor necrosis factor alpha; NASH, Nonalcoholic steatohepatitis; TGF-β, transforming growth factor-β.

## Data Availability

Not applicable.

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
