# Peer review of "Metformin Prevents or Delays the Development and Progression of Osteoarthritis: New Insight and Mechanism of Action"

_cells, 2022, doi:10.3390/cells11193012_

Round 1

Reviewer 1 Report

The Review title is “Metformin Prevents or Delays the Development and Progression of Osteoarthritis: New Insight and Mechanism of Action.” In this review, the authors aimed to summarize the latest research progress on metformin in OA. In this article, the authors explained all aspects of the AMPK pathway, SIRT1 expression, and metformin combination therapy based on a comprehensive analysis. This well-written review covered all available clinical studies. Therefore, the authors concluded that metformin can be effective in preventing or delaying the onset of OA. In this review, the author has written well, and the argument is strong enough to convince me. This is a review that I consider suitable for publication.

Author Response

Reply: Thank you very much for your kindly comments!

Reviewer 2 Report

The review by He et al. summarizes the current state of the knowledge around metformin and its non-primary therapeutic usage, the recently proposed and introduced treatment of osteoarthritis. While the manuscript is well structured, it misses some substantial insights into the mechanisms at the molecular levels. Authors speak about the therapies on humans and mention the effects of bovine serum albumin; it would be more relevant to mention works done with the human serum albumin and its interactions with metformin and also more complex interplays. Metformin action is also dependent on metal ions, which was demonstrated already. This fact is, of course, not without meaning for osteoarthritis, where metal ions' levels are essential and play a role. Therefore I would like to ask to expand the metformin and metal ions discussion and mention more about the cross-talk with the human serum albumin.

The following references will be helpful and please cite them:

https://pubmed.ncbi.nlm.nih.gov/31153112/

https://pubmed.ncbi.nlm.nih.gov/24720899/

https://pubmed.ncbi.nlm.nih.gov/31184536/

https://pubmed.ncbi.nlm.nih.gov/34443319/

https://pubmed.ncbi.nlm.nih.gov/34204936/

https://pubmed.ncbi.nlm.nih.gov/35898971/

https://www.biorxiv.org/content/10.1101/2022.07.04.498643v1

I am convinced that metformin cross-talks with these functions of albumin described above and has the ability to coordinate selected metal ions, influencing the body's fluid homeostasis.

Overall, I would also recommend checking English for typos and spaces; sometimes there are too many.

Author Response

Reply: Thank you very much for your suggestions! We have added the metformin cross-talks as wells as modified the language by a professional company.

Reviewer 3 Report

The authors described metformin effects associated with OA.

The manuscript is not readable, contains many flaws. It is scientifically illiterate.

Comments

1.      The language of the manuscript requires improvement in scientific writing using the help of professional editing company such as American Journal Experts (AJE), Wiley Editing Services etc. In addition, the authors should choose Advanced Editing option.

2.      The manuscript contains many contradictory statements. For example:

Line 70 versus line 81

Line 100 versus line 111

Line 372 versus line 373

Line 272 versus line 273 etc.

3.      Line 33: Ref #17 declares that metformin only could “limit OA development and progression in injury-induced OA animal models”. This should be corrected.

4.      Section 2: The authors should describe the structure of metformin.

5.      Lines 97-99: this information is not important for he Review article and it should be removed.

6.      In the Review article the authors should analyze previously published data, therefore, all the statements should be supplied by corresponding references. References are required on lines 119, 154, 158,180,189,195,208, 212,316, etc

7.      All the repeated statements should be removed:

Line 198 and line 202;

Lines 54-55 and 300-302 etc.

8.      Many sentences are not clear: Lines 123-127; 135-137,184-187, etc.

9.      Some terms are not clear. For instance, area units? This should be corrected.

10.  Table 1 should be supplemented with some examples of studies related to metformin effects on human OA.

11.  The manuscript contains misleading information. For example, lines 351-357 do not correspond to the information described in ref # 95, etc.

12.  Conclusion: Lines 417-418 are not evident from the manuscript. This should be corrected.

13.  Section 3: The description of OA pathogenesis is doubtful as its description is messy. No references were presented. This should be corrected.

Author Response

The authors described metformin effects associated with OA.

The manuscript is not readable, contains many flaws. It is scientifically illiterate.

Comments

  1. The language of the manuscript requires improvement in scientific writing using the help of professional editing company such as American Journal Experts (AJE), Wiley Editing Services etc. In addition, the authors should choose Advanced Editing option.

Reply: Thank you very much for your suggestions! The language has been edited by a professional editing company with certificate in attachment.

  1. The manuscript contains many contradictory statements. For example:

Line 70 versus line 81

Line 100 versus line 111

Line 372 versus line 373

Line 272 versus line 273 etc.

Reply: Thank you very much for your suggestions! We have corrected these statements.

  1. Line 33: Ref #17 declares that metformin only could “limit OA development and progression in injury-induced OA animal models”. This should be corrected.

Reply: Corrected.

  1. Section 2: The authors should describe the structure of metformin.

Reply: Thank you very much for your kindly suggestions! The structure of metformin has been added in Section 2.

  1. Lines 97-99: this information is not important for the Review article and it should be removed.

Reply: This part has been removed.

  1. In the Review article the authors should analyze previously published data, therefore, all the statements should be supplied by corresponding references. References are required on lines 119, 154, 158,180,189,195,208, 212,316, etc

Reply: Thank you very much for your suggestions! We double checked these sentence and find we actually have references for them. The reference are always placed at the end of each case.

  1. All the repeated statements should be removed:

Line 198 and line 202;

Lines 54-55 and 300-302 etc.

Reply: This part has been removed.

  1. Many sentences are not clear: Lines 123-127; 135-137,184-187, etc.

Reply: Thank you very much for your suggestions! These sentences have been refined.

  1. Some terms are not clear. For instance, area units? This should be corrected.

Reply: These sentences have been modified.

  1. Table 1 should be supplemented with some examples of studies related to metformin effects on human OA.

Reply: Thank you very much for your suggestions! We have added these studies.

  1. The manuscript contains misleading information. For example, lines 351-357 do not correspond to the information described in ref # 95, etc.

Reply: Sorry about that. ref # 95 and related content has been removed.

  1. Conclusion: Lines 417-418 are not evident from the manuscript. This should be corrected.

Reply: Thank you very much! We have modified this part.

  1. Section 3: The description of OA pathogenesis is doubtful as its description is messy. No references were presented.This should be corrected.

Reply: Thank you very much for your suggestions! This part does not add new information in the field and therefore has been removed.

Round 2

Reviewer 3 Report

The review remains rather messy, contains many repeats, many sentences are not clear and meaningful, the language is not good in spite of the presented Certificate.

M=metformin

Comments

1.      These sentences are not clear. They should be clarified:

Line 28: “M can significantly treat diabetes”.???

Lines 33-34: The authors should indicate of OA is prevented by M?

Lines 73-74: It is not clear, how drugs can be synthesized via disease mechanisms?

Line 83-84: This sentence should be rephrased.

Line 93: “separation of oxidative phosphorylation” should be explained.

Line 97: “restrictive property of M on mitochondrial injury” should be explained.

2.      Lines 58-61: These statements are contradictory. The authors should explain and resolve this contradiction.

3.       Lines 61-65: Logical connection is missing. This should be corrected.

4.      Lines 107-111: This fragment should be presented in more detail and logically connected with previous section.

5.      Lines 83-88 should be placed after Line 117.

6.      Lines 91-93 and 144-146, 146-148 contain contradictory statements. This should be corrected.

7.      Lines 51-53 repeat lines 152-153. This should be corrected.

8.      Lines 5o repeats lines 168-169. This should be corrected.

9.      Lines 180-181: This sentence is not associated with previous description. Lines 51-53 repeat lines 152-153. This should be corrected.

10.  Lines 221-222: This sentence is not clear. This should be corrected.

11.  Lines 257-258: This sentence is not clear. Reference is required at theenmd of the sentence. This should be corrected.

12.  Line 352: It is not clear which “metformin proteins” the authors describe.This should be corrected.

13.  Lines 358-361: It is not clear why intraarticular injection is not considered an invasive process? This should be clarified.

Author Response

The review remains rather messy, contains many repeats, many sentences are not clear and meaningful, the language is not good in spite of the presented Certificate.

M=metformin

Comments

  1. These sentences are not clear. They should be clarified:

Line 28: “M can significantly treat diabetes”.???

Lines 33-34: The authors should indicate of OA is prevented by M?

Lines 73-74: It is not clear, how drugs can be synthesized via disease mechanisms?

Line 83-84: This sentence should be rephrased.

Line 93: “separation of oxidative phosphorylation” should be explained.

Line 97: “restrictive property of M on mitochondrial injury” should be explained.

Reply: Sorry for the inaccurate expression! We have modified or explained these sentences.

  1. Lines 58-61: These statements are contradictory. The authors should explain and resolve this contradiction.

Reply: Sorry for the confusion! We have reorganized this part.

  1. Lines 61-65: Logical connection is missing. This should be corrected.

Reply: Thank you very much for your kindly suggestions! We have corrected.

  1. Lines 107-111: This fragment should be presented in more detail and logically connected with previous section.

Reply: Sorry for the confusion! We found this part not relate to the topic and it has been removed.

  1. Lines 83-88 should be placed after Line 117.

Reply: Thank you very much for your kindly suggestions! We have corrected.

  1. Lines 91-93 and 144-146, 146-148 contain contradictory statements. This should be corrected.

Reply: Thank you very much for your kindly suggestions! We have corrected.

  1. Lines 51-53 repeat lines 152-153. This should be corrected.

Reply: Thank you very much for your suggestions! Actually, the line you're referring to are inconsistent with what I've seen, and I've tried my best to find what you're referring to and make corrections. If I didn’t correct the right place, please point out the specific sentence. Thank you very much!

  1. Lines 5o repeats lines 168-169. This should be corrected.

Reply: Thank you very much for your suggestions! Corrected.

  1. Lines 180-181: This sentence is not associated with previous description. Lines 51-53 repeat lines 152-153. This should be corrected.

Reply: Thank you very much for your suggestions! Corrected.

  1. Lines 221-222: This sentence is not clear. This should be corrected.

Reply: Thank you very much for your suggestions! Corrected.

  1. Lines 257-258: This sentence is not clear. Reference is required at theend of the sentence. This should be corrected.

Reply: Thank you very much for your suggestions! Corrected.

  1. Line 352: It is not clear which “metformin proteins” the authors describe.This should be corrected.

Reply: Thank you very much for your suggestions! Corrected.

  1. Lines 358-361: It is not clear why intraarticular injection is not considered an invasive process? This should be clarified.

Reply: Thank you very much for your suggestions! Corrected.